# Improving Transformer-based Program Repair Models through False Behavior Diagnosis

**Youngkyoung Kim**
Department of Electrical and
Computer Engineering,
Sungkyunkwan University
agnes66@skku.edu

**Misoo Kim**
Department of Artificial
Intelligence Convergence,
Chonnam National University
misoo.kim@jnu.ac.kr

**Eunseok Lee**[*]
College of Computing
and Informatics,
Sungkyunkwan University
leees@skku.edu

## Abstract

Research on automated program repairs using transformer-based models has recently gained considerable attention. The comprehension of the erroneous behavior of a model enables the identification of its inherent capacity and provides insights for improvement. However, the current landscape of research on program repair models lacks an investigation of their false behavior. Thus, we propose a methodology for diagnosing and treating the false behaviors of transformer-based program repair models. Specifically, we propose 1) a behavior vector that quantifies the behavior of the model when it generates an output, 2) a behavior discriminator (BeDisc) that identifies false behaviors, and 3) two methods for false behavior treatment. Through a large-scale experiment on 55,562 instances employing four datasets and three models, the BeDisc exhibited a balanced accuracy of 86.6% for false behavior classification. The first treatment, namely, early abortion, successfully eliminated 60.4% of false behavior while preserving 97.4% repair accuracy. Furthermore, the second treatment, namely, masked bypassing, resulted in an average improvement of 40.5% in the top-1 repair accuracy. These experimental results demonstrated the importance of investigating false behaviors in program repair models.

## 1 Introduction

Automated program repair (APR), a technique that has attracted substantial attention from both academia and industry, can mitigate the costs associated with bug fixing during software development and maintenance by automatically generating fixing patches. Transformers have received enormous attention and have been applied to APR, showing effectiveness in fixing defects, bugs, errors, and vulnerabilities [1] (Fu et al., 2022; Pearce et al., 2022; Ahmad et al., 2021; Mashhadi and Hemmati, 2021; Prenner and Robbes, 2021).

Because existing APR studies have primarily focused on developing and improving the APR model, the internal decision-making process behind patch generation is often overlooked. Investigating the inner workings of a model can offer valuable insights by identifying its capabilities and limitations (Mohammadkhani et al., 2023; Palacio et al., 2023; Borowski et al., 2020; Gunning and Aha, 2019; Voita et al., 2019; Ribeiro et al., 2016a,b; Katuwal and Chen, 2016). Specifically, false behaviors of the model can provide a direction for improvement. For example, Kim et al. improved bug localization accuracy by eliminating the input tokens that the model considers when it fails to localize (Kim et al., 2022c).

Similarly, an examination of the internal behavior of the model during the generation of incorrect patches can provide direction for improvement. Current studies on incorrect patches primarily focus on distinguishing incorrect patches by relying primarily on the embedding vectors of the input buggy code (Phung et al., 2022; Csuvik et al., 2020). However, because of the repeated reformulation of these vectors within the inner layers of the transformer-based model, the embedding vector lacks sufficient information to discern the process that causes incorrect patch generation. Moreover, once an incorrect patch is identified, determining directions to improve becomes challenging. Therefore, research must be conducted with a focus on identifying and addressing false inner workings (that is, false behaviors), in which the model generates incorrect patches.

Applying the methods and findings of previous studies that analyzed the model behavior may appear like an intuitive solution. However, directly utilizing them becomes challenging because of the disparity between our target task, program repair, and the tasks investigated in previous studies. Code

---

[*]corresponding author

[1]We refer to these as bugs in a broad sense.

fixing is a generation task that poses difficulties when the results or methodologies typically used in classification tasks are applied (Kim et al., 2022c; Ribeiro et al., 2016a). Among the various generation tasks (Voita et al., 2019), code fixing focuses specifically on eliminating bugs from the source code.

To ensure consistency in the input/output language and preserve the correct structure and semantics of the code, the majority of the inputs must remain unchanged. Furthermore, as most existing studies focus on comprehending the successful or overall behaviors of a model (Rabin et al., 2021; Voita et al., 2019; Ribeiro et al., 2016a), a deficiency exists in understanding false behaviors and identifying their characteristics for automatic detection. Therefore, it is essential to conduct research to comprehend false behaviors, along with their identification and subsequent treatment for the APR domain.

This study proposes a methodology for diagnosing and treating false behaviors in transformer-based program repair models. We propose a behavior vector to represent the internal behavior of the model. The proposed behavior vector is generated by extracting the attention weights and value vectors from each attention head inside the transformer model. This approach enables us to capture the model behavior in terms of the way it considers or neglects the input tokens during the patch generation steps. Additionally, we introduce a behavior discriminator (BeDisc) that distinguishes the patterns between successful and false behaviors exhibited by the model. Based on this, we introduce two treatments 1) early abortion and 2) masked bypassing. Our contributions are as follows:

- We introduce a novel perspective for improving program repair tasks by analyzing the internal behavior of a transformer-based program repair model.

- We present a methodology for representing the internal behavior of a transformer-based program repair model and propose an approach for diagnosing and mitigating false behaviors exhibited by models for the APR task.

- The results of a large-scale experiment on 55,562 instances using seven pairs of models and datasets showed average identification rates of 86.9% for true patches and 86.4% for incorrect patches by a BeDisc.

- Our first treatment filtered 60.4% of the incorrect patches, significantly reducing the patch verification time. Additionally, our second treatment, namely, masked bypassing, increased the generation of correct patches by a minimum of 5.8% and up to 130.4%.

## 2 Preliminaries and Related Works

### 2.1 Attentions in Transformer Architecture

Attention heads are the core elements of the transformer architecture (Hao et al., 2021; Wang et al., 2020; Voita et al., 2019). The transformer extensively utilizes an attention mechanism, representing the input tokens in various context vectors. Context vectors are used to compute the likelihood of each vocabulary as the next token. In a transformer model, each attention head $AH^{lh}$ ($h$-th head of $l$-th layer) reformulates a context vector based on the tokens in input string $S = \{x_1, x_2, ..., x_n\}$. In the first layer (block), embedding vector $X$ is projected onto the query, key, and value spaces, with projection weight $W^Q, W^K, W^V$, respectively, as shown in Equation 1. Notably, previous studies on incorrect patches have employed the initial embedding vector $X$ before projection to evaluate patch correctness.

$$Q, K, V = XW^Q, XW^K, XW^V \qquad (1)$$

$Q$ and $K$ are used to calculate attention map $A$ that stores the relevancies between tokens. In Attention is all you need, where the transformer is first proposed, relevancies are calculated with a scaled dot product as in Equation 2 (Vaswani et al., 2017). They proposed to scale the dot product value by $\sqrt{d_k}$, where $d_k$ represents the size of the key vector, to prevent the dot product value from becoming excessively large.

$$A = Softmax(\frac{QK^\top}{\sqrt{d_k}}) \qquad (2)$$

The relevance of tokens $x_i$ and $x_j$ is stored as $A_{ij}$ in matrix $A$, which is real number $\alpha_{ij} \in \mathbb{R}$ denoting the attention that $x_i$ pays to $x_j$. Owing to the use of multiple attention heads with distinct weight matrices $W^Q, W^K, W^V$ in each $AH^{lh}$, relevance in diverse perspectives is stored in each map $A$ of the $AH^{lh}$. Finally, context vector matrix $C$ is derived by applying Equation 3. Consequently, the context vector corresponding to each token $x_i$ can be expressed as shown in Equation 4 by decomposing Equation 3 with respect to $C_i$.

$$C = AV \tag{3}$$

$$C_i = \Sigma_{j=0}^{n} \alpha_{ij} V_j \tag{4}$$

Within an attention block, multiple $AH^{lh}$s concurrently compute context vector matrix $C^{lh}$, which is subsequently treated as a new embedding vector and propagated through the feed-forward layer. The reformulation process is repetitively iterated in successive blocks. The behavior of the model is influenced by these $AH^{lh}$s. The $AH^{lh}$s make the model focus on different parts of the input sequence for each token, considering their relevance from various perspectives. In this study, we propose a behavior vector to quantify model behavior by analyzing the focus on target token $x_t$ during the internal processes of each $AH^{lh}$.

## 2.2 Transformer for Program Repair

The effectiveness of the transformer-based program repair model has been experimentally demonstrated in both encoder-decoder families [2] (Li et al., 2022; Kim et al., 2022b; Wang et al., 2021; Berabi et al., 2021) and decoder-only families(Jesse et al., 2023; Joshi et al., 2022; Prenner and Robbes, 2021), with their correct patch generation accuracy. The program repair model is trained to transform the input buggy code into a fixed code (that is, a patch). The generated patches are verified through manual or test case evaluations to filter out incorrect patches. The developers apply a patch that successfully passes the verification.

In recent studies, repair model such as TFix, demonstrated the repair ability of T5-based repair model on 52 error types of JSLint(Berabi et al., 2021), and Fu et al. utilized the T5 model for vulnerability repair (Fu et al., 2022). PLBART fixed 19.21% and 8.98% of bugs, respectively, in the small and medium datasets of `Patches in wild` (Ahmad et al., 2021). CodeT5 fixed 22.59% and 14.18% of the bugs, respectively, for the same datasets by incorporating a code-specific tokenizer (Wang et al., 2021). Mashhadi et al. showed that CodeBERT could fix simple bugs in Java with an accuracy of $19 \sim 72\%$ (Mashhadi and Hemmati, 2021). Kim et al. empirically demonstrated 20% bug-fixing accuracy for Kotlin bugs in an industrial environment using TFix (Kim et al., 2022b). Codex successfully resolved 23 bugs out of 40

[2]https://huggingface.co/docs/transformers/model_summary

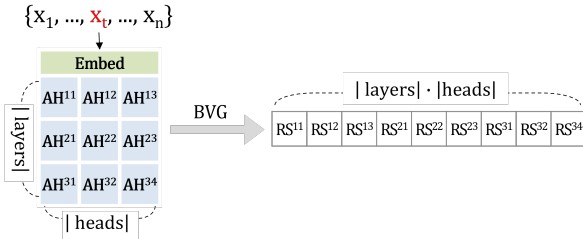

Figure 1: Illustration of generating behavior vector representation for the target token.

QuixBug benchmarks (Prenner and Robbes, 2021) and was able to address security vulnerabilities (Pearce et al., 2022). Overall, transformers have exhibited promising results for program repair. However, improving the model without accounting for its behavior during patch generation may result in ineffective improvements (Mohammadkhani et al., 2023).

## 2.3 Model Behavior: Existing Approaches

To understand and interpret the behavior of the model, researchers in the fields of NLP and software engineering have explored explainable artificial intelligence (XAI), specifically focusing on the model consideration of input tokens and the influential role of specific neurons in determining the model output(Jiarpakdee et al., 2020; Wattanakriengkrai et al., 2020; Rabin et al., 2021; Suneja et al., 2021). The attention weight, a common medium for interpreting the behavior of deep learning models with attention mechanisms, provides insight into how the model refers to specific tokens (Li et al., 2016). Prior studies on program repair models demonstrated their code-fixing ability by utilizing attention maps, highlighting the focused words during patch generation(Lutellier et al., 2020, 2019; Jiang et al., 2021). However, simply tracking the attention weight may be insufficient to capture the model behavior. This limitation arises from neglecting the size of the token vectors and misaligning their actual impacts (Kobayashi et al., 2020). In Equation 4, the effect of the target token $x_t$ when calculating $C_i$ is related to attention weight $\alpha_{it}$ and vector $V_t$ of token $x_t$. Considering all these factors, we propose a behavior vector that indicates the reference level of the model for each word. By leveraging this vector, we can identify false behaviors and apply further treatments.

# 3 Methodology

## 3.1 Behavior Vector Representation

We propose a behavior vector $b_t \in \mathbb{R}^{|layers| \cdot |heads|}$ that represents the internal process of the transformer model on a buggy input token $x_t$ when generating a fixing code (that is, a patch). The proposed behavior vector reflects the extent to which the model considers or neglects token $x_t$ during the generation of patch $S_{out} = \{x_{n+1}, x_{n+2}, ..., x_m\}$ for an input with buggy code $S_{in} = \{x_1, x_2, ..., x_n\}$.

Figure 1 illustrates the proposed methodology to obtain the behavior vector for target token $x_t$ from a transformer-based repair model with a behavior vector generator (BVG). To generate behavior vector $b_t$, the BVG iterates each $AH^{lh}$ and calculates the relative stake ($RS^{lh}$) of $x_t$ in computing the context vectors used to predict an output token at each generation step. In the subsequent description, note that the target token is referred to as $x_t$, the input tokens are denoted as $x_j$, and the output tokens are designated as $x_i$.

Recalling Equation 4, calculating context vector $C_i$ for token $x_i$ in each $AH^{lh}$ can be interpreted as a weighted sum of tokens $\forall x_j \in S_{in}$. In other words, each input token $x_j$ has a different stake in calculating $C_i$ for generation, and the values that determine this stake are attention weight $\alpha_{ij}$ and size of the value vector $\|V_j\|$. Hence, the stake of target token $x_t$ in calculating $\forall C_i \in C$ in each $AH^{lh}$ can be represented by Equation 5.

$$stake(x_t) = \Sigma_{t=0}^{|C|} \alpha_{it} \|V_t\| \qquad (5)$$

The relative stake of model reference to target token $x_t$ compared with other input tokens $\forall x_j \in S_{in}$ during generation, denoted $RS^{lh}$ in Figure 1, is calculated as Equation 6.

$$RS^{lh} = \frac{\Sigma_{i=0}^{|C|} \alpha_{it} \|V_t\|}{\Sigma_{j=0}^{|S_{in}|} \Sigma_{i=0}^{|C|} \alpha_{ij} \|V_j\|} \qquad (6)$$

For calculation, we collect the Euclidean norm of value vectors for every $x_j$ in $AH^{lh}$. We process attention maps at each generation step to construct an attention map $A^{lh} \in \mathbb{R}^{|S_{out}| \times |S_{in}|}$ where attention weights $\alpha_{ij}$ for every $x_i$ and $x_j$ are stored. From each generation step where token $x_i$ is selected as the output token, we can extract the attention weight that each $x_j$ received from token $x_i$ and concatenate them as $\mathbf{a} \in \mathbb{R}^{|S_{in}|}$. Then, every $\mathbf{a}$ from

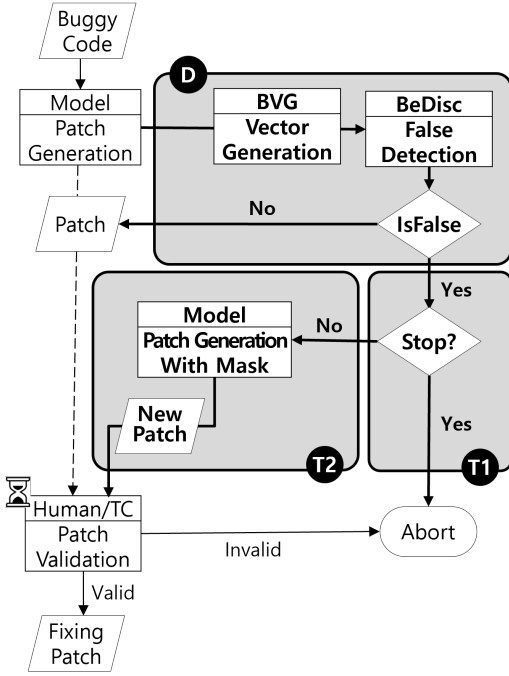

Figure 2: Program repair with false behavior diagnosis and treatment.

the generation step is concatenated into the final attention map $A^{lh} \in \mathbb{R}^{|S_{out}| \times |S_{in}|}$.

Consequently, a scalar value $RS^{lh}$ representing the manner each $AH^{lh}$ considers $x_t$ during the generation step can be obtained. The obtained $RS^{lh}$ values for each $AH^{lh}$ are then concatenated into behavior vector $b_t \in \mathbb{R}^{|layers| \cdot |heads|}$ of the model for target token $x_t$. The behavior vector for a multi-token set $S_a \subseteq S_{in}$ can be represented in two ways. First, to represent the behavior in a fine-grained manner, $b_t$ for all tokens can be concatenated as $B_t \in \mathbb{R}^{|S_a| \times |layers| \cdot |heads|}$. Second, the vectors of all tokens, $\forall x_t \in S_a$, can be averaged and used as a behavior vector to capture the behavior of the model on tokens of the specific type as a collective entity.

## 3.2 Program Repair with False Behavior Diagnosis and Treatment

Figure 2 shows the program repair process using the proposed false behavior diagnosis and treatment. The uncolored portion in the figure corresponds to the typical program repair method described in Section 2.2. Instead of directly passing the model-generated patch to the subsequent validation step (indicated by the dotted line), which can be a significant bottleneck, the proposed method incorporates the process of diagnosing and treating false behaviors, as depicted in the gray boxes.

Treatment methods T1 and T2 can be selected based on the development environment.

**False Behavior Diagnosis.** The box labeled *D* in Figure 2 represents the diagnosis of the false behaviors when the model generates a patch. The BeDisc is a binary classifier that uses the behavior vector (BV) of the model as input and determines whether it corresponds to a behavior that could result in generating an incorrect patch (that is, a false behavior). The generated patch is returned if the BeDisc predicts that the behavior is true. Otherwise, the next treatment step follows. The validation (or training) data used to train (or validate) the patch generation model can be used to train the BeDisc. The label of the behavior vector is assigned a value of 1 if the repair model successfully generates a correct patch for the input buggy code; otherwise, it is set to 0.

**Treatment 1: Abortion.** An effective treatment for false behaviors is early abortion, as denoted by the *T1* box in Figure 2. This method immediately aborts the patch before the validation stage. After generating a patch, the APR model undergoes a validation process to assess the patch. This can be conducted manually by the developer or by employing a test suite to verify the compliance with all test cases. Manual validation can be laborious, and test suites can incur significant costs to build and compile the entire software to execute test cases. Therefore, aborting an incorrect patch without proceeding to the validation stage can save considerable time.

**Treatment 2: Masked Bypassing.** This method prevents the model from referencing a suspicious target token that might have caused the false behavior when generating the patch. We use an attention mask to assign a value of zero to the target tokens, thereby avoiding their influence. The attention mask can indicate the model in which tokens should be attended to (Hugging Face, 2020; Kim et al., 2022a). For example, when we want to disregard the token $b$ for a token sequence $\{a\ b\ c\}$, we can use the masking array $\{1, 0, 1\}$. Then, based on the given masking array, the model will use $\{a$ _ $c\}$ as the input instance. To bypass the effect of the target token without altering the positional information of the remaining tokens, we use an attention mask instead of replacing the token with alternative tokens such as <MASK>.

Treatment 1 is recommended for software with numerous test cases or large-scale systems to streamline the validation process. Otherwise, Treat-

| | | | | | Test | | Validation | |
|---|---|---|---|---|---|---|---|---|
| Idx | L | H | Model | Data | #Input | #Fix | #Input | #Fix |
| 1 | 6 | 8 | TFixS | JSLint | 10,504 | 4,670 | 9,454 | 4,245 |
| 2 | 12 | 12 | TFixB | JSLint | 10,504 | 5,004 | 9,454 | 4,480 |
| 3 | 12 | 12 | CT5 | WS | 5,835 | 1,246 | 5,835 | 1,253 |
| 4 | 12 | 12 | CT5 | WM | 6,545 | 777 | 6,546 | 765 |
| 5 | 12 | 12 | CT5 | WS$^{na}$ | 5,835 | 1,513 | 5,835 | 1,424 |
| 6 | 20 | 16 | CGen | WS | 5,835 | 853 | 5,835 | 755 |
| 7 | 20 | 16 | CGen | JSLint | 10,504 | 3,258 | 9,454 | 2,937 |
| | | Total | | | 55,562 | 17,321 | 52,413 | 15,859 |

Table 1: Experimental datasets and models. **TFixS** indicates TFix-small, **TFixB** indicates TFix-base, **CT5** indicates CodeT5-base, **CGen** indicates CodeGen-350M-multi, **WS** indicates Wild-small, **WM** indicates Wild-medium. Validation data of the APR model are used to train the BeDisc.

ment 2 could be preferable for maximizing the number of correct patches, prioritizing repair accuracy over saving validation time.

## 4 Experimental Setup

### 4.1 Models and Datasets

We conducted experiments using four models (TFix-small, TFix-base, CodeT5-base, and CodeGen-350M-multi) on three publicly available datasets (Table 1). Pairs 1-4 were validated in previous studies (Berabi et al., 2021; Wang et al., 2021), whereas pairs 5-7 were added for our experiments. To analyze the effect of the proposed method with and without token abstraction, we introduced the fifth pair, Wild-small$^{na}$, using the Wild-small dataset without abstraction. The sixth and seventh pairs involved the CodeGen model (Nijkamp et al., 2022), which is designed specifically for software-related text generation. The $|layers|$ and $|heads|$ of each model are listed in columns L and H. The #Input and #Fix columns represent number of inputs and the number of inputs exhibiting true behavior (correct fixing code generated as top-1), respectively. Instances with false behaviors outnumbered those with true behaviors. Input and output for each model are available in Appendix A.2.

### 4.2 Metrics

**BeDisc Accuracy.** To evaluate the BeDisc, we used classification evaluation metrics including the F1 score, sensitivity (true positive rate; TPR), specificity (true negative rate; TNR), and balanced accuracy (Brodersen et al., 2010) where higher values indicate better performance.

**Top-k Fixing Accuracy.** To evaluate the repair performance, we examined the top-k generated

| Idx | neuron | thres | TPR | TNR | F1 | Bal_Acc |
|-----|--------|-------|-------|-------|-------|---------|
| 1 | 420 | 0.64 | **94.87** | 70.78 | 82.28 | 82.82 |
| 2 | 410 | 0.61 | 90.26 | 80.64 | **85.54** | 85.45 |
| 3 | 440 | 0.67 | 86.65 | 91.13 | 80.86 | 88.89 |
| 4 | 460 | 0.84 | 89.16 | **95.22** | 81.56 | **92.19** |
| 5 | 480 | 0.60 | 80.34 | 90.79 | 78.78 | 85.57 |
| 6 | 430 | 0.66 | 80.76 | 87.94 | 67.34 | 84.35 |
| 7 | 390 | 0.60 | 85.90 | 88.08 | 80.95 | 86.99 |
| | Average | | 86.85 | 86.37 | 79.62 | 86.61 |

Table 2: Diagnosis accuracy of the BeDisc with the proposed behavior vector. The best performance of each metric is marked in bold.

patches for each instance and considered them to be a successful fix if any patch within the top-k was correct. In accordance with previous studies (Wang et al., 2021; Berabi et al., 2021; Kim et al., 2022b), an exact match with the developer's patch was considered as a correct patch. For $k > 1$, an effort to determine the first correct patch was evaluated using a mean reciprocal rank (MRR).

### 4.3 Hyperparameters

We employed a single hidden layer MLP as BeDisc to assess the performance achievable using a simple classifier. We selected need-fix token sets among the input tokens and used the average vector as the final behavior vector; these are crucial tokens among the buggy codes for fixing. We evaluated the results on the test dataset using the hyperparameters obtained from a three-fold validation process. The hyperparameter setting with the highest average value across the three-fold validation was selected. The model was then trained on the entire validation dataset and used to make inferences on the test dataset to obtain the final results. Further details on hyperparameters are available in Appendix A.1.

## 5 Results

### 5.1 Accuracy of Behavior Discriminator

Table 2 lists the classification performances of the proposed BeDisc. We selected the hyperparameters with F1 score. In the experimental data, a threshold of 0.6 or higher was selected for all pairs because those with false behaviors outnumbered instances with true behaviors. The neuron size of the optimal model can indicate the complexity of the target task because larger models have more expressive capacity (Hu et al., 2021). The selected neuron range (390~480) was relatively large within the hyperparameter search space, suggesting that learning behavioral patterns is a non-trivial task. Fur-

ther studies are required to effectively distinguish false behavior patterns. Nevertheless, the proposed single-layer BeDisc achieved a classification accuracy of 86.61% by balanced accuracy (Bal_Acc) and 79.61% by F1 score on average, surpassing 50% Bal_Acc of the random classifier.

The first pair, which showed the second-best F1 score and best TPR, had the worst TNR and Bal_Acc. Overall, a higher TPR corresponded to a lower TNR performance. Classifiers with a high TNR can better filter the false behavior but might result in degradation in the fixing ability. Therefore, the hyperparameters for the classifier should be selected based on the development environment, considering factors such as the software scale.

The BeDisc demonstrated its effectiveness in both the encoder-decoder and decoder-only models, albeit with certain distinctions. When comparing the model architectures, the order of performance based on the F1 score was TFix > CodeT5 > CodeGen, whereas that based on the Bal_Acc standard was CodeT5 > CodeGen > TFix. When comparing the two pairs with the same wild-small dataset but different architecture (the third and sixth pairs), CodeT5 exhibited better classification accuracy across all four evaluation metrics. However, for the same JSLint dataset (the first, second, and seventh pairs), CodeGen exhibited better Bal_Acc.

The behavior vector size for the first and second pairs, TFix-small and TFix-base, were 48 and 144, respectively, with TFix-base being three times larger than TFix-small. However, they displayed only a marginal difference of approximately 3p% for both F1 and Bal_Acc. The seventh pair, which utilized the JSLint dataset, had a behavior vector size of 320, exceeding TFix-small by more than six times. However, the difference in the performance was less than 5p%. The results indicated that the BeDisc could effectively discriminate false behaviors using the proposed behavior vectors across transformer models of different sizes.

The third and fifth were pairs with and without text abstraction on the wild-small dataset, and the BeDisc classified better for code that performed abstraction than for raw code with a marginal difference. The sixth and seventh pairs used different data on the same model. The JSLint data that had more instances for the BeDisc to learn, performed better than the wild-small data. This trend remained when comparing the first and second pairs that used JSLint and the third–fifth pairs that

| | | Intended Tolerance for Search | | | | |
|---|---|---|---|---|---|---|
| | | -1% | -2% | -3% | -4% | -5% |
| 1 | RealTol | -2.2 | -2.8 | -4.2 | -9.2 | -11.7 |
| | TNR | 61.1 | 67.0 | 71.0 | 78.7 | 81.1 |
| 2 | RealTol | -1.5 | -3.1 | -4.8 | -6.4 | -7.5 |
| | TNR | 51.5 | 56.3 | 68.9 | 72.2 | 75.0 |
| 3 | RealTol | -0.9 | -1.5 | -0.8 | -2.6 | -3.1 |
| | TNR | 61.6 | 62.6 | 58.9 | 74.0 | 74.4 |
| 4 | RealTol | -0.6 | -3.0 | -2.2 | -5.5 | -6.9 |
| | TNR | 48.9 | 82.0 | 81.7 | 90.5 | 91.5 |
| 5 | RealTol | -1.4 | -4.3 | -5.1 | -7.0 | -13.0 |
| | TNR | 42.6 | 69.9 | 73.2 | 78.0 | 85.9 |
| 6 | RealTol | -2.0 | -2.4 | -2.9 | -12.6 | -9.7 |
| | TNR | 35.7 | 42.7 | 50.2 | 72.8 | 72.8 |
| 7 | RealTol | -1.1 | -1.1 | -2.7 | -10.6 | -12.8 |
| | TNR | 39.1 | 42.0 | 56.7 | 74.9 | 79.9 |
| Average | RealTol | -1.4 | -2.6 | -3.2 | -7.7 | -9.2 |
| | TNR | 48.6 | 60.4 | 65.8 | 77.3 | 80.1 |

Table 3: Early abortion performance by different false negative tolerances.

used the Wild dataset.

## 5.2 T1: Early Abortion

This section presents the results of the first treatment, namely, early abortion. We selected the hyperparameters based on the highest FNR of the validation dataset for each tolerance, aiming to maximize the early abortion of false behaviors while meeting the degradation criteria. Therefore, we examined the effectiveness of the early abortion treatment in achieving accurate abortion while tolerating performance degradation within the range of -1% to -5%. Table 3 lists the obtained experimental results.

Disparities can exist between the intended tolerance and the real tolerance (RealTol), that indicates the performance deterioration rate. Despite performance degradation (-7.7% and -9.2%), BeDiscs successfully aborted a substantial portion of false behaviors, with success rates of 77.3% and 80.1% for tolerance levels of -4% and -5%, respectively. When aiming for a tolerance range of -1% to -3%, only a slight additional performance degradation (average of -0.4p% and maximum of -2.3p%) occurred than intended and 48.6% to 65.8% of false behaviors were successfully aborted. Additionally, between -3% and -4%, a sharp increase was noted in the performance degradation. In summary, we recommended setting the tolerance for hyperparameter search to -4% to -5% for large-scale software that requires significant time for building and testing to minimize the time spent on validating

| | | OG | w/o BeDisc, w Mask | | w BeDisc, w Mask | |
|---|---|---|---|---|---|---|
| | #Input | #Fix | #Fix | imp | #Fix | Imp |
| 1 | 10,504 | 4,670 | 3,062 | -34.4% | **5,017** | 7.4% |
| 2 | 10,504 | 5,004 | 1,748 | -65.1% | **5,272** | 5.4% |
| 3 | 5,835 | 1,246 | 871 | -30.1% | **1,543** | 23.8% |
| 4 | 6,545 | 777 | 1,640 | 111.1% | **1,790** | 130.4% |
| 5 | 5,835 | 1,513 | 1,271 | -16.0% | **1,900** | 25.6% |
| 6 | 5,835 | 853 | 1,438 | 68.6% | **1,559** | 82.8% |
| 7 | 10,504 | 3,258 | 2,407 | -26.1% | **3,469** | 6.5% |
| Total | 55,562 | 17,321 | 12,437 | -28.2% | **20,550** | 18.6% |

Table 4: Fixing accuracy with and without the proposed treatment, highlighting the best repair accuracy for each pair in bold.

| | Intended Tolerance for Search | | | | |
|---|---|---|---|---|---|
| | -1% | -2% | -3% | -4% | -5% |
| 1 | 8.4% | **8.9%** | 8.7% | 7.9% | 7.4% |
| 2 | **5.8%** | 5.2% | **5.8%** | 5.5% | 5.4% |
| 3 | 19.2% | 19.0% | 18.1% | **23.8%** | **23.8%** |
| 4 | 74.1% | 117.9% | 116.2% | 128.6% | **130.4%** |
| 5 | 16.2% | 24.9% | 26.0% | **26.6%** | 25.6% |
| 6 | 46.1% | 49.1% | 58.6% | **84.3%** | 82.8% |
| 7 | 6.9% | 7.1% | **7.9%** | 6.7% | 6.5% |
| Avg. | 25.2% | 33.2% | 34.5% | **40.5%** | 40.3% |

Table 5: Repair performance improvement regarding BeDisc hyperparameter search tolerance. The best improvement for each model is presented in bold.

incorrect patches. Conversely, we recommend a tolerance range of -1% to -3% for the opposite scenario.

## 5.3 T2: Masked Bypassing

### 5.3.1 Top-1 Accuracy

This section presents the results of the second treatment, that is, masked bypassing. We used an identical BeDisc from the -5% column in Table 3. Table 4 compares the fixing accuracy of the original model (OG column), the proposed method (w BeDisc, w Mask column), and an ablation case in which the target tokens are masked irrespective of the false behavior judgment (w/o BeDisc, w Mask column) of the model.

The proposed treatment method (w BeDisc, w Mask) resulted in an average improvement of 18.6% in the repair accuracy across all seven pairs, with the best case generating 130.4% more correct patches. In contrast, when masking was applied to all instances without considering false behaviors (w/o BeDisc, w Mask), the performance deteriorated in five of the seven combinations, resulting in an average decrease of -28.2%. Compared with the proposed method, a -65.2% ($\approx \frac{(20,550-12,437) \times 100}{12,437}$) decrease was observed in re-

|       | MRR        |       |      | Ranks in Original Model |     |     |     |     |       |     |
| :---: | :--------: | :---: | :--: | :---: | :-: | :-: | :-: | :-: | :---: | :-: |
|       | OG | OG+T2 | Imp. | 1 | 2 | 3 | 4 | 5 | 6~50 | N/A |
| 1     | 0.50 | 0.53 | 7%  | 2  | 109 | 33  | 32  | 13 | 174 | 164 |
| 2     | 0.52 | 0.55 | 5%  | 0  | 80  | 21  | 13  | 5  | 97  | 155 |
| 3     | 0.31 | 0.35 | 12% | 0  | 36  | 18  | 5   | 8  | 108 | 104 |
| 4     | 0.19 | 0.27 | 39% | 0  | 74  | 33  | 36  | 26 | 298 | 207 |
| 5     | 0.32 | 0.36 | 13% | 0  | 20  | 13  | 8   | 4  | 74  | 167 |
| 6     | 0.27 | 0.32 | 19% | 63 | 30  | 22  | 12  | 8  | 143 | 157 |
| Total | 0.35 | 0.40 | 16% | 65 | 349 | 140 | 106 | 64 | 894 | 954 |

Table 6: Investigation result of false behaviors with top-50 accuracies. N/A denotes the scenario in which the correct patch is not found among the generated patches.

pair accuracy because the ability of the proposed method to assess true/false behaviors on target tokens enabled their preservation when they led to correct patch generation.

The repair performance of CodeGen (the sixth pair) improved from 853 to 1,559 on the wild-small dataset, whereas that of CodeT5 (the third pair) improved from 1,246 to 1,543. Before the proposed treatment, CodeT5 showed a better repair accuracy; however, with the proposed treatment, CodeGen could fix more defects. The target token caused more false behaviors in CodeT5 models than in CodeGen, which is a more recent model. Even without BeDisc (w/o BeDisc, w Mask), 68.6% more fixes were possible in the sixth pair. Further investigation is required because the type of input that leads to significant false behaviors can vary across models.

Table 5 presents the fixing accuracy improvement rate corresponding to BeDiscs with different tolerance. For each tolerance, the same model shown in Table 3 was used. Irrespective of the tolerance we selected for the hyperparameter search, the second treatment improved the repair accuracy. Across the first, second, and seventh pairs, for each model using the JSLint dataset, the variation in improvement between the least and the most was less than 2p%. Pairs using the wild dataset exhibited greater variation, yet consistently achieved optimal performance with tolerances of -4% and -5%.

### 5.3.2 Top-k Accuracy

We assessed the ranking of correct patches within the top-k generated patches of the model. This evaluation demonstrated the efficacy of the proposed treatment method and highlighted the negative impact of false behaviors on accurate patch generation. Considering the capacity of the experimental environment, we set k as 50 and used a beam size of 50. The seventh pair was excluded because of the capacity limitations of the experimental environment.

This pair incorporates the largest model and text input, posing a computational capacity issue when generating top-50 patches with a beam size of 50. This caused the model to fail in some cases due to an out-of-memory error, making it unsuitable for a comprehensive comparison. In this section, we used a BeDisc with a tolerance of 1%.

We compared the effort required to find the first correct patch with (OG+T2 column) and without (OG column) the proposed treatment, using the MRR metric. In contrast to OG, where developers receive the initial top-50 patches from the original model, OG+T2 employs a subsequent process to restructure the patch rankings. Following the generation by the original model, the model's behavior is evaluated using BeDisc. If the behavior is determined erroneous (i.e., when the current top-1 patch is identified as incorrect), two steps are taken. In the first step, a patch is generated using masked attention, employing T2's treatment. Then, in the second step, the original top-1 patch is replaced with the top-1 patch from T2. On the other hand, if the behavior is determined as accurate (i.e., the current top-1 patch is considered correct), no changes are made to the top-50 patches, and they remain unchanged.

The results are presented in the MRR columns of Table 6. The OG+T2 column shows the MRR of the top-50 ranked patch list following the application of treatment T2. This evaluates the efficiency of finding a correct fixed patch within the ranked patches. A higher MRR indicates a quicker identification of the correct fixed patch, consequently reducing the effort required for validation. The higher the MRR, the less effort is required until the developer encounters the correct fixed patch. For example, if the correct patch is ranked 10th, validation of the preceding nine patches is necessary. Conversely, if the correct patch is ranked top-1, no effort will be consumed for validation. The Imp. column denotes the performance improvement rate of OG+T2 compared with that of OG. Top-50 repair attempts on the APR model revealed that the original model achieved an MRR of 0.35. However, by avoiding false behaviors, a 16% higher MRR was achieved. This improvement was achieved by lowering the ranks of incorrect patches, facilitating the prompt identification of correct patches. The experimental results shed light on the impact of false behaviors on patch generation.

To conduct a thorough analysis, we examined

2,572 instances that were identified as false behaviors but could be fixed to generate the correct patch as top-1 by applying the second treatment. We investigated the rankings of the corresponding patches in the original model.A small number of the examined instances ($3\% \approx \frac{65 \times 100}{2572}$) were initially generated with the correct patch as the top-1 but were classified as false behavior owing to judgment errors in the BeDisc. However, the majority ($97\% \approx \frac{2507 \times 100}{2572}$) were cases where the patch could not be generated as the top-1 owing to false behaviors.

Specifically, $72\%$ ($\approx \frac{(894+954) \times 100}{2572}$) of the instances that attained the correct patch as the top-1 result after applying the proposed treatment were initially unable to generate the correct patch within the top-5 ranks owing to false behavior in the original model. Additionally, 50% of them, which corresponded to 894 instances, failed to generate the correct patch even when the search progressed up to the top-50 patches. This suggested that false behaviors could significantly hinder the ability of a model to sort the necessary vocabulary for generating correct patches. Given that the proposed treatment method simply aims to mitigate the impact of false behaviors, further refinement of the false behavior treatment method is expected to enable a higher rate of correct patch generation.

## 6 Conclusion

This study is the first to examine the inner workings of program repair model failures and to introduce a behavior vector for diagnosing false behaviors in a transformer-based program repair model. We proposed methods for diagnosing and addressing the false behavior of the program repair model and validated their effectiveness across three models and four datasets. The proposed BeDisc and treatment strategies improved the APR performance by 18.6% on average and up to 130.4%. The experimental results demonstrated the necessity for additional investigations into the false behavior of program repair models. In future studies, we intend to expand the proposed method to a wider range of models and bugs. In Addition, we intend to investigate false behaviors across broader target tokens to develop an advanced treatment approach.

## Limitations

**Repair Model.** We experimented with three representative models, excluding the encoder-only models owing to their sub-optimal nature (Wang et al., 2021). Although certain repair models may not have been trained with optimal hyperparameters, we used the CodeT5 and TFix models with the same performance as in previous studies, suggesting the anticipated effectiveness of the proposed method in typical cases.

**BeDisc Architecture.** The BeDisc used in the experiment may not be the best architecture for identifying false behaviors. However, despite its simplicity, the model identified 60.4% of false behaviors with -2.6% of tolerance. Further studies on advanced model architectures for identifying false behaviors can lead to additional improvements in the BeDisc.

**Tolerance Setting.** The selection of appropriate tolerance is a human-driven process because it may vary across developmental environments, potentially leading to challenges in determining tolerance settings. However, because we confirmed that the proposed method was consistently effective within a tolerance of -1% to -5%, we recommended deciding based on the experimental findings.

**Behavior diagnosis target token type.** Accurately identifying the location of the target token (that is, need-fix token) can pose challenges, particularly in the present experiment, in which we assumed perfect localization of the need-fix token. However, this can be supplemented by techniques that locate fixed locus (Phung and Lee, 2020; Chandra et al., 2011). We plan to explore a wider range of tokens to address this issue in future studies.

## Acknowledgements

This work was supported by the National Research Foundation of Korea(NRF) grant funded by the Korea government(MSIT)(2019R1A2C2006411).

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

## A Implementation Detail

The code and data used for our experiment can be found in the supplement. We modified Transformers 4.12.3[3] to extract the normalized attention map. The experiments were conducted with one GPU (GeForce RTX 3090) with 24 GB RAM.

### A.1 Hyperparameter Settings for training Models

**BeDisc.** The hyperparameters for BeDisc included a neuron range of 0 to 500 and a class determination threshold range of 0.01 to 1. We used the default settings of scikit-learn [4] for other hyperparameters. It is important to note that hyperparameters are searched using validation data and subsequently applied to test data. Consequently, the performance of the proposed method can be further enhanced by utilizing optimized hyperparameters specific to the test dataset.

**APR Models.** We followed previous studies (Chen et al., 2023; Alexandr et al., 2021; Khandelwal et al., 2019) to train the CodeGen model and fine-tuned it for one epoch because large language models (LLMs) such as CodeGen are susceptible to overfitting (Jiang et al., 2023; Fried et al., 2022). We trained CodeGen-350M-multi [5] with a learning rate of 5e-5 and batch size 4, with a max length of 512. The pretrained tokenizer does not have `pad_token`, so we have set `eos_token` as `pad_token`. The repair model is trained with a Transformers library [3]. In the JSLint dataset, there were 84,846 instances for training, 9,454 for validation, and 10,504 for the test set. For the WS dataset, the corresponding numbers were 46,680 for training, 5,835 for validation, and 5,835 for the test set. The default values of PyTorch[6] were used for unspecified hyperparameters. For the CodeT5 model, the pre-trained CodeT5 model is fine-tuned with small and medium datasets of `Patches in wild` (Ahmad et al., 2021), independently with a learning rate initialized to $10^{-4}$ with batch size 32. TFix was trained with a learning rate initialized to $10^{-4}$ with batch size 32. The pre-trained T5-base model is trained with ESLint javascript defects.

---

[3]https://pypi.org/project/transformers/4.12.3/
[4]https://scikit-learn.org/0.24/
[5]https://huggingface.co/Salesforce/codegen-350M-multi
[6]https://pytorch.org/docs/1.7.1/

> **Input:**
> `fix` no-undef Undefined variable. z = obj.foo();
> `:`
> bar(5);
> z = obj.foo();
> **Output:**
> bar(5);
> var z = obj.foo();

Figure 3: Input and output format of the TFix model

### A.2 Prompt and Post-process

Special tokens (e.g., eos, sos tokens) were excluded from the string output of the model and employed as the final output for evaluation. The input/output settings for each model are as follows:

**TFix.** The JSLint dataset used in the TFix model followed the prompt template of the original study. As shown in Figure 3, the datasets trained the model to output fixed code for the input in the form " fix {rule id} {error message} {buggy line} :\n {buggy code} ".

**CodeT5.** Wild datasets use buggy code as input, with no additional prefixes or postfixes. Unlike TFix, dataset, the dataset used only buggy code as an input to output fixed code .

**CodeGen.** The CodeGen model is a decoder-only model trained for causal language modeling, predicting the next token in a sequence. Therefore, fine-tuning was performed to learn the sequences of "{original input format of the dataset} Fixed: {fixed code} ". For inference, the *original input* was provided to the model to generate the text. The text returned by the model was split with "Fixed:" and posterior tokens were treated as a generated patch.

## B Effectiveness of Treatments in Mitigating False Negatives of BeDisc

False negatives of BeDisc are the cases where the repair model's correct behavior is wrongly classified as false behavior. As our experimental results in Table 2, whose hyperparameter search criteria was F1-score, a false negative rate (FNR) of classification was 13.63% ($\approx 100 - 86.37$). This diminishes the efficacy of the repair model by wrongly discarding the correct patch. We introduced a *Intended Tolerance for Search* to address this issue. By modifying the classifier hyperparameter search criteria from F1-score to best TNR within speci-

| Pair | Impact on Repair Model | -1% | -2% | -3% | -4% | -5% |
|------|------------------------|-----|-----|-----|-----|-----|
| | *Wrong Rejection (FN)* | 95 | 123 | 180 | 398 | 504 |
| 1 | Additional Fix ($T2_{FN}$) | 39 | 60 | 98 | 243 | 322 |
| | Additional Fix ($T2_{TN}$) | 449 | 477 | 486 | 522 | 529 |
| | *Wrong Rejection (FN)* | 18 | 56 | 65 | 90 | 167 |
| 5 | Additional Fix ($T2_{FN}$) | 2 | 16 | 21 | 33 | 63 |
| | Additional Fix ($T2_{TN}$) | 261 | 417 | 437 | 460 | 490 |
| | *Wrong Rejection (FN)* | 37 | 35 | 87 | 344 | 416 |
| 7 | Additional Fix ($T2_{FN}$) | 14 | 17 | 43 | 174 | 212 |
| | Additional Fix ($T2_{TN}$) | 248 | 249 | 303 | 388 | 415 |

Table 7: Effectiveness of T2 recovering the negative impact of FNs on the repair model.

| Vector Type | F1 | Bal_Acc | TPR | TNR |
|-------------|------|---------|--------|--------|
| Last token | 43.171 | 59.324 | 77.076 | 41.572 |
| Avg of all tokens | 43.17 | 59.71 | 74.04 | 45.39 |
| Avg of NeedFix tokens | 52.36 | 67.25 | 66.53 | 67.96 |
| Behavior Vector (Ours) | 78.78 | 85.57 | 80.34 | 90.79 |

Table 8: Classification accuracy comparison with other vector representations.

fied *Intended Tolerance for Search*, we can find a classifier that rejects correct patches within an endurable range. As a result, our treatment T1 was able to reject 48.6% to 80.1% of false behaviors within -1.4% to -9.2% of repair accuracy (corresponding to the average row in Table 3). Hence, T1 offers significant time savings for developers with a minor trade-off. Given the subjective nature of a "minor trade-off", developers have the flexibility to establish an *intended tolerance* in accordance with their requirements.

Regarding our second treatment (T2), both TNs and FNs of the classifier trigger the repair model to regenerate patches with masking. Regenerating the patch for FNs may potentially compromise repair accuracy as their correct patch has already been generated. Nonetheless, T2 was able to recover the negative impacts of FNs by successfully generating accurate patches for both FNs and TNs. We present specific numbers and impacts on the repair model across the 1st, 5th, and 7th pairs (one pair per model) in Table 7. For the *Intended tolerance for search* setting ranging from -1% to -5%, the first rows of each pair show the number of wrong rejections due to FNs. However, as shown in subsequent rows, our proposed treatment T2 enabled substantial additional fixes for both FNs ($T2_{FN}$) and TNs ($T2_{TN}$). Notably, the combined number of $T2_{TN}$ and $T2_{FN}$ surpassed the number of FNs, resulting in an overall improvement of the repair model.

## C Comparing Vector Representations for Behavior Classification

For the purpose of behavior classification, employing the input/output vector (e.g., embedding vector, context vector) of the model itself is feasible. However, our behavior vector offers an advantage in terms of explicitness and compactness, resulting in improved classification accuracy.

**Explicitness.** Token vectors are the input/result of inner workings, while our behavior vector encapsulates the process of inner workings. For instance, context vector $C_i$ for a token $x_i$ in each $AH^{lh}$, is a weighted sum of tokens $\forall x_j \in S_{in}$ (please refer to Equation 3 and Equation 4 in Section 2). Such tokens may capture contextual information of input tokens, yet they cannot express how the model handles the tokens. This inherent ambiguity poses challenges in distinguishing between the correct and incorrect approaches the model adopts for token handling. However, our behavior vector offers a quantitative measure of the model's utilization of each token during its internal computations, consequently providing a more explicit representation of the model's behavior.

**Compactness.** Our behavior vector depicts the internal model behavior in a layer-wise and head-wise manner. Achieving a similar representation using token vectors significantly inflates the vector size to $|layers| \times |heads| \times embedding\_size$. For instance, CodeT5-base requires a vector size of $110,593 (= 12 \times 12 \times 768)$, whereas our approach entails a vector size of just $144 (= 12 \times 12)$. Utilizing only the final context vector of the encoder could reduce redundancy, but it would result in significantly lower explicitness.

In table 8, we report additional experimental results on the CodeT5-base model with the $WS^{na}$ dataset to compare the effectiveness of behavior vectors against token vectors for false behavior classification. We employed the final context vector(last hidden state of the encoder) of the last token, which was either ";" or "}". Specifically, we selected the token that consistently appears in the same location. Therefore, the last token, which was either ";" or "}" was chosen. Hyperparameters for the classifier were selected with validation data that maximized the F1-score, which was the same method we used in Table 2. The classification accuracy was 43.17 of F1-score and 59.32 of Bal_Acc, which are 47% ($\approx \frac{80.86-43.17}{80.86}$) and 33% ($\approx \frac{88.89-59.32}{88.89}$) lower than our behavior vector. For further comparison, we have conducted

experiments using the average vector of all tokens, which is another way to retrieve sentence embedding (Choi et al., 2021; Huang et al., 2021). However, it resulted in lower accuracy; 43.17 for F1 and 59.71 for Bal_Acc. Additionally, we have conducted experiments using NeedFix tokens, resulting in 52.36 for F1 and 67.25 for Bal_Acc. The third and fourth rows in Table 8 summarizes the results. These results show the need for a novel approach to detect false behaviors of the model upon token vector embedding.