# OpenReview forum: "Improving Transformer-based Program Repair Model through False Behavior Diagnosis"
_EMNLP/2023/Conference — EMNLP 2023 Main_

### Official Review · Reviewer_GLov · 2023-08-02

**Typos Grammar Style And Presentation Improvements:** In line 12 of section 2.1, "wight" sh…
**Soundness:** 3

**Excitement:**

3: Ambivalent: It has merits (e.g., it reports state-of-the-art results, the idea is nice), but there are key weaknesses (e.g., it describes incremental work), and it can significantly benefit from another round of revision. However, I won't object to accepting it if my co-reviewers champion it.

**Paper Topic And Main Contributions:**

- This paper explores the diagnosis and treatment of False Behavior in the Patches generation process within the Program Repair Model.
- This paper proposes a highly compatible framework for diagnosing and treating False Behavior and accordingly introduces the Behavior Vector for diagnosis. Additionally, two methods for treatment are proposed: abortion and masked bypassing.
- Experimental results on numerous datasets show that simply adding the modules proposed in this paper can effectively increase the number of correctly repaired programs, without changing the baseline.


**Questions For The Authors:**

- Is the "Intended Tolerance for Search" in Table 5 derived under the same parameter settings as the "Intended Tolerance for Search" in Table 3?


**Reasons To Accept:**

This paper enhances the performance of the program repair model by analyzing its internal generation process from a novel perspective. This direction of improvement represents a promising path worth exploring.

**Reasons To Reject:**

- The writing and readability of this paper leave room for improvement. Some key points in the article lack detailed descriptions and explanations, making the content unclear and difficult for readers to comprehend.
    - The paper fails to clearly explain the difference between the Attention MASK introduced in section 3.2. There is a lack of example illustrations.
    - The meanings of each row and column in the results tables are not clearly explained, and the purposes of some experiments are not explicitly stated. Furthermore, it is unclear whether the parameters of the models remain consistent across different experimental results. It would be beneficial to clarify whether the observed improvements result from parameter variations or methodological enhancements.
    - The distinctions between Balanced Accuracy and the F1 score are not clearly explained, nor are the respective functions of these two metrics.


**Reproducibility:**

3: Could reproduce the results with some difficulty. The settings of parameters are underspecified or subjectively determined; the training/evaluation data are not widely available.

**Reviewer Confidence:**

4: Quite sure. I tried to check the important points carefully. It's unlikely, though conceivable, that I missed something that should affect my ratings.

---

> ### Author Rebuttal · Authors · 2023-08-28
>
> Thank you for the valuable comments regarding the readability of our paper. In response to the reviewer's comments, we will enhance the paper's clarity by providing more specific explanations. Below is the response to the reviewer's question.
>
> __Q1. Is the "Intended Tolerance for Search" in Table 5 derived under the same parameter settings as the "Intended Tolerance for Search" in Table 3?__
>
> Yes, the results in Tables 3 and 5 utilized the same hyperparameters for each tolerance.  We apologize for the inconsistent column names, which could have led to confusion. We will modify "Intended Tolerance" in Table 3 to "Intended Tolerance for Search" to align with Table 5.
>
> The BeDisc's hyperparameters were selected based on validation data from the dataset for each pair. Therefore, the same hyperparameters are selected as long as the pair and "intended tolerance" are identical. We introduced the concept of "Intended Tolerance" to confine FNs to a limited scope, thus preventing the rejection of accurate patches. The classifier's FNs can erroneously discard correct patches, thereby reducing program repair accuracy. To address this, we searched hyperparameters that maximize the TNR (False patch correctly identified as False) while maintaining program repair accuracy within a specified standard.
>
> __Concern about description on attention mask:__
> For further clarification, we would like to add an explanation about the attention mask.
>
> > Attention mask can indicate the model in which tokens should be attended to [1][2]. For example, when we want to disregard the token *b* for a token sequence {*a b c*}, we can use the masking array {1, 0, 1}, and the model uses {*a _ c*} as the new input instance based on this masking array.
>
> [1] https://huggingface.co/transformers/v3.5.1/glossary.html#attention-mask
> [2] Impact of Defect Instances for Successful Deep Learning-based Automatic Program Repair (ICSME'22)

---

### Official Review · Reviewer_aJHn · 2023-08-04

**Soundness:** 4

**Excitement:**

4: Strong: This paper deepens the understanding of some phenomenon or lowers the barriers to an existing research direction.

**Paper Topic And Main Contributions:**

**Summary:**

Current Automatic Program Repair (APR) studies mainly focus on improving these models to generate correct patches for the faulty program. However, investigating the false behavior of these models was not addressed in the previous works. To address this, the paper proposes a methodology for diagnosing and treating the false behaviors of the transformer-based program repair models. The paper mainly proposes a behavior vector together with a behavior discriminator to quantify the behavior of the models and employ them to improve the general APR models’ performance.

**Contributions:**

-The paper introduces a novel perspective to improve program repair tasks by diagnosing and treating the false behaviors of transformer-based program repair models.
-The paper proposes a behavior vector based on transformer-based models to diagnose and mitigate false behaviors of the model in generating the patches for the given buggy program.
-The evaluation results for 55,562 instances of different datasets showed the effectiveness of the approach in identifying correct and incorrect patches (It correctly classified 86.9% of correct and 86.4% of incorrect patches).


**Questions For The Authors:**

- It is not clear if the proposed behavior vector is the best option to provide representation for classifying the patches. Could you please justify why we can not use the model itself to classify the patches? Or adding a special token (e.g., [CLS] token) to the inputs and employing that as the behavior vector representation?
- It is not clear how the CodeGen model was trained (fine-tuned). Could you please provide the training details, including the learning rate, input structure, and input size?
- Do you think the output of BeDisc can be used as an implicit uncertainty estimation?
- Table 6 is a bit vague. What does T2 stand for (I know it refers to the treatments. However, it would be better to provide the details)? Why does the 7th pair cause the computational capacity issue?


**Reasons To Accept:**

- The paper proposed a novel perspective to improve program repair tasks by considering the false behavior of the models.
- It provides extensive experimental results on three different models and four different datasets.
- It shows the effectiveness of the approach in identifying correct and incorrect patches. Furthermore, it employs false behavior diagnoses to improve the APR models.
- The paper is well-written and easy to follow.


**Reasons To Reject:**

- The novelty of the behavior vector is questionable and it is not clear if it works better than using the model itself to classify the patches (e.g., using a specific token like [CLS] as the classification vector representation, see the “Questions For The Authors”) .
- The training details of the CodeGen model are not provided. E.g., the input size, learning rate, etc.


**Reproducibility:**

3: Could reproduce the results with some difficulty. The settings of parameters are underspecified or subjectively determined; the training/evaluation data are not widely available.

**Reviewer Confidence:**

3: Pretty sure, but there's a chance I missed something. Although I have a good feel for this area in general, I did not carefully check the paper's details, e.g., the math, experimental design, or novelty.

**Typos Grammar Style And Presentation Improvements:**

- At 2.1 add what is d_k (Eq 2)? I suggest to provide one or two sentences about that.

---

> ### Author Rebuttal · Authors · 2023-08-28
>
> We appreciate the insightful comments and acknowledgment of our main contribution. If we get the chance to submit the final version of the paper, we will include supplementary details in the appendix. Here, we address the concerns and questions raised.
>
> __Q1.  Could you please justify why we can not use the model itself to classify the patches? Or adding a special token (e.g., [CLS] token) to the inputs and employing that as the behavior vector representation?__
> Employing the model itself based on the vector (e.g., embedding vector, context vector) of a specific token for the purpose of patch classification is feasible. However, our behavior vector is more explicit yet compact in depicting the model's behavior.
>
>  - Explicitness: Token vectors are the *input/result* of inner workings, while our behavior vector encapsulates the *process* of inner workings. For instance, the context vector of a target token results from contextual adjustment involving a weighted vector sum of input tokens. However, we cannot know *how* it was used for contextual adjustment. This inherent ambiguity renders it less explicit than our behavior vector, which estimates the degree to which the model utilized the token during context adjustment. Thus, our behavior vector offers a more explicit representation of the inner workings, making it easier to identify the false behavior of the model.
>
> - Compactness: Our behavior vector depicts the internal model behavior in a layer-wise and head-wise manner. Achieving a similar representation using token vectors significantly inflates the vector size to |layers| * |heads| * embedding size. For instance, CodeT5-base requires a vector size of 110,593 (12 * 12 * 768), whereas our approach entails a vector size of just 144 (12 * 12). Utilizing only the final context vector of the encoder could reduce redundancy, but it would result in significantly lower explicitness.
>
> Also, we have conducted additional experiments on the CodeT5-base model with the wild-small-na dataset to compare the effectiveness of behavior vectors against token vectors for false patch classification. We employed the final context vector(last hidden state of the encoder) of the last token, which was either ";" or "}". Hyperparameters for the classifier were selected with validation data that maximized the F1-score, which was the same method we used in Table 2. The classification accuracy was 43.17 of F1-score and 59.32 of Bal_Acc, which are 47% (=(80.86-43.17)/80.86) and 33% (=(88.89-59.32)/88.89) lower than our behavior vector.
>
> __Q2. It is not clear how the CodeGen model was trained. Could you please provide the training details, including the learning rate, input structure, and input size?__
> For input and output structure we have described it in the last paragraph of Appendix A.2.
> >...fine-tuning was performed to learn the sequences of"{original input format of the dataset} Fixed: {fixed code}". For inference, the original input was provided to the model to generate the text. The text returned by the model was split with “Fixed:” and posterior tokens were treated as a generated patch.
>
> We trained CodeGen-350M-multi [1] with a learning rate of 5e-5 and batch size 4, with a max length of 512. The pretrained tokenizer does not have pad_token, so we have set eos_token as pad_token. The repair model is trained with a transformers library[2]. The specific modeling code of Codegen we used for our experiments can be found in *"Supplements\transformers.zip\transformers\models\codegen"*. In the JSLint dataset, there were 84,846 instances for training, 9,454 for validation, and 10,504 for the test set. For the WS dataset, the corresponding numbers were 46,680 for training, 5,835 for validation, and 5,835 for the test set.
>
>
> __Q3. Do you think the output of BeDisc can be used as an implicit uncertainty estimation?__
> This question is intriguing. Since the behavior vector represents how the model regard/disregard the token, using it for uncertainty estimation appears feasible. We should explore how model uncertainty relates to 1) the Deviation of RS^{lh} across layers and heads, in addition to 2) the overall value of RS^{lh} value for uncertainty estimation. Although uncertainty estimation is not our expertise, we speculate that behavior vectors could aid in estimating token-level uncertainty in language models. We would like to further investigate this topic in the future.
>
>
> __Q4. Table 6 is a bit vague. What does T2 stand for? Why does the 7th pair cause the computational capacity issue?__
> The objective of this experiment, reported in Table 6, was to investigate how severe the false behavior of the original model was, which can be treated by T2. If T2 is applicable to only moderate false behaviors, it will lift the rank of patches that were already in high rank to Top1. Also, we wanted to check if the remedy is possible by simply enlarging the beam size. However, our investigation shows that T2 was applicable to severe false behaviors, whose fixing was impossible in the Top-50, even with enlarged beam size. The "Ranks in Original Model" column shows the number of instances that T2(with beam-size 5 generation) can improve its rank from x-th-> 1st (x<=2). The MRR column shows its impact overall. The higher the MRR, the less effort is required to validate the patch. The MRR improvement was possible because T2 was able to move the rank of the correct patch to Top-1, whose rank was even out of Top-50 (with beam-size 50 generation).
>
> The 7th pair incorporates the largest model and the largest text input. This pair caused the computational capacity issue when generating top-50 patches with beam size 50, which is the minimal beam size we need to generate top-50 patches. Consequently, the model failed to generate patches in some instances because of an OOM error, even with inference batch size one. Therefore, we did not investigate this pair as the result might be an incomplete comparison.
>
> [1] https://huggingface.co/Salesforce/codegen-350M-multi
> [2] Transformers: State-of-the-Art Natural Language Processing (https://github.com/huggingface/transformers)

---

### Official Review · Reviewer_1EGD · 2023-08-12

**Soundness:** 3

**Excitement:**

4: Strong: This paper deepens the understanding of some phenomenon or lowers the barriers to an existing research direction.

**Paper Topic And Main Contributions:**

Automated program repair is an widely discussed topic and has potential to reduce costs associated with bug fixing in software development
and maintenance procedure. This paper proposes some novel techniques to improve performance of such program repairs.
Authors proposed a method to diagnose and treat false behaviors of transformer-based program repair models with the help of a behavior vector, a behavior discriminator (BeDisc) that identifies false behaviors, and two methods for false behavior treatment.
Their novelty is in improving program repair tasks by analyzing the internal behaviors of transformer-based models by demonstrating internal behavior of a transformer-based program repair model.

**Questions For The Authors:**

A. Can you please clarify impact of false negatives on the system?

**Reasons To Accept:**

Along with a good overall presentation of the paper it has some strengths. Treatment 1, Abortions has good potential false behavior diagnosis. Also Treatment 2, masked Bypassing eliminates suspicious target tokens responsible for false behaviour while producing patches.
Overall improvements seem to be significant. Qualitative and quantitatively significant datasets are used.

**Reasons To Reject:**

One of the weakness is that as per table 2. still there are some false negatives. It is not clear how significant can such false negatives be for such systems.
Another weak point is overall the paper is not easy to read for general audience.

**Reproducibility:**

4: Could mostly reproduce the results, but there may be some variation because of sample variance or minor variations in their interpretation of the protocol or method.

**Reviewer Confidence:**

3: Pretty sure, but there's a chance I missed something. Although I have a good feel for this area in general, I did not carefully check the paper's details, e.g., the math, experimental design, or novelty.

---

> ### Author Rebuttal · Authors · 2023-08-28
>
> We want to thank the reviewer for acknowledging our main contribution. As for concerns about comprehensibility for a broader audience, we will enhance clarity by providing more detailed explanations, including supplementary details in the appendix. Below, we address the raised question.
>
> __Q1. Can you please clarify impact of false negatives on the system?__
> False negatives of BeDisc are the cases where the repair model's correct behavior is wrongly classified as false behavior. As indicated by the reviewer's feedback, Table 2, whose hyperparameter search criteria was F1-score, displays a false negative rate (FNR) of 13.63% (100% - 86.37%) in the classification outcomes. This makes the system discard the correct patch, diminishing the efficacy of the repair model. In our paper, we addressed this by modifying the classifier's hyperparameter search criteria from F1-score to TNR with "Intended Tolerance". Therefore, we use the hyperparameter that maximizes false behavior detection within the specified "Intended Tolerance", which is determined by FNR. This adjustment reduces FNs and prevents the rejection of correct patches.
>
> As a result, our treatment T1 was able to reject 48.6% ~ 80.1% of false behaviors with -1.4% ~ -9.2% of repair accuracy (corresponding to the average row in Table 3). Hence, T1 offers significant time savings for developers with a minor trade-off. Given the subjective nature of a "minor trade-off," developers have the flexibility to establish an "intended tolerance" in accordance with their requirements.
>
> Regarding our second treatment (T2), negatives (Ns) of the classifier trigger the repair model to regenerate patches using masking. This occurs even if the correct patch has been generated without masking (FNs), potentially causing inefficiencies. Nonetheless, T2 successfully generates accurate patches for both FNs and true negatives (TNs). We present specific numbers and impacts on the repair model across the 1st, 5th, and 7th pairs (one pair per model) in Table 1. For the "Intended tolerance for search" setting ranging from -1% to -5%, the initial rows of each pair show the count of false negatives (FNs). This might reduce the repair model's accuracy due to incorrect rejections. However, T2 enables substantial additional fixes for both FNs (T2_FN) and TNs (T2_TNs). Notably, the combined count of T2_TN and T2_FN surpasses the number of FNs, resulting in overall improvement of the repair model.
>
>
> Table 1. Impact of False Negatives on Repair Model and Impact of T2 Despite the Presence of FNs.
> | Pair | `Impact on Repair Model` | -1% | -2% | -3% | -4% | -5% |
> |:---:|:---:|:---:|:---:|:---:|:---:|:---:|
> | 1 | *Wrong Rejection (FN)* | 95 | 123 | 180 | 398 | 504 |
> |  | Additional Fix (T2_FN) | 39 | 60 | 98 | 243 | 322 |
> |  | Additional Fix (T2_TN) | 449 | 477 | 486 | 522 | 529 |
> | 5 | *Wrong Rejection (FN)* | 18 | 56 | 65 | 90 | 167 |
> |  | Additional Fix (T2_FN) | 2 | 16 | 21 | 33 | 63 |
> |  | Additional Fix (T2_TN) | 261 | 417 | 437 | 460 | 490 |
> | 7 | *Wrong Rejection (FN)* | 37 | 35 | 87 | 344 | 416 |
> |  | Additional Fix (T2_FN) | 14 | 17 | 43 | 174 | 212 |
> |  | Additional Fix (T2_TN) | 248 | 249 | 303 | 388 | 415 |

---

### Meta-Review · Area_Chair_zZCP · 2023-09-01

**Recommendation:** Accept (Poster)
**Confidence:** 5

**Metareview:**

Program behavior diagnosis is a very challenging and important problem, and the authors have made an interesting and impactful contribution to the problem space. While their approach has false positives and negatives, these seem comparable even to some static analysis methods I have seen for detecting bugs.

---

### Decision · Program_Chairs · 2023-10-07

**Decision:**

Accept-Main

**Comment:**

Program behavior diagnosis is a very challenging and important problem, and the authors have made an interesting and impactful contribution to the problem space. While their approach has false positives and negatives, these seem comparable even to some static analysis methods I have seen for detecting bugs.